# Understanding and Stabilizing GANs' Training Dynamics with Control Theory

## Abstract

Generative adversarial networks (GANs) have made significant progress on realistic image generation but often suffer from instability during the training process. Most previous analyses mainly focus on the equilibrium that GANs achieve, whereas a gap exists between such theoretical analyses and practical implementations, where it is the training dynamics that play a vital role in the convergence and stability of GANs. In this paper, we directly model the dynamics of GANs and adopt the control theory to understand and stabilize it. Specifically, we interpret the training process of various GANs as certain types of dynamics in a unified perspective of control theory, which enables us to model the stability and convergence easily. Borrowed from control theory, we adopt the widely-used negative feedback control to stabilize the training dynamics, which can be considered as an $L2$ regularization on the output of the discriminator. We empirically verify our method on both synthetic data and natural image datasets. The results demonstrate that our method can efficiently stabilize the dynamics and converge to better results.

## 1 Introduction

Generative adversarial networks (GANs) (Goodfellow et al., 2014) have shown promise in generating realistic natural images (Brock et al., 2018) and facilitating unsupervised and semi-supervised learning (Chen et al., 2016; Li et al., 2017; Donahue & Simonyan, 2019). In GANs, an implicit generator ($G$) is defined by mapping a noise distribution to the data space. Since no density function is defined for the implicit generator, a discriminator ($D$) is introduced to estimate the density ratio between the data distribution $p_D$ and the generating distribution $p_G$ by telling the real samples from fake ones. $G$ aims to recover the data distribution by maximizing this ratio. This framework is formulated as a minimax optimization problem, which can be solved by optimizing $G$ and $D$ alternately. In practice, however, GANs can be extremely sensitive to hyperparameters (Lucic et al., 2018; Radford et al., 2015), and oscillations are often observed (Liang et al., 2018; Chvardova & Fleuret, 2018), thereby suffering from the instability of training (Goodfellow, 2016).

There are some theoretical analyses aiming to understand and stabilize the training process of GANs, such as designing different objective functions using other statistical divergences (Nowozin et al., 2016; Nguyen et al., 2017; Du et al., 2018; Mao et al., 2017), and introducing auxiliary regularization terms (Gulrajani et al., 2017; Arjovsky et al., 2017). However, there is a gap between the theoretical analyses and practical implementations: most of previous work assumes that (1) $D$ achieves optimal when training $G$ (Goodfellow et al., 2014; Arjovsky et al., 2017; Gulrajani et al., 2017; Nowozin et al., 2016); and (2) the optimal $D$ is a smooth function of $G$ (Metz et al., 2016). These two assumptions are violated in most practical cases. First, GANs are optimized by alternating between $k$ steps of optimizing $D$ and one step of optimizing $G$, where $k$ generally takes 1 or 5. It results in a sub-optimal $D$, especially in the early stage of the training process. Second, $D$ is constant w.r.t. $G$ when $G$ is optimized, which results in missing gradients and an unstable training process (Metz et al., 2016).

Instead, to ensure practical convergence, it is the training dynamics of both $G$ and $D$, i.e., how $G$ (or $D$) changes given current $G$ and $D$ during the training process, that should be considered, where the previous two assumptions are no more required. There are some methods that focus on the dynamics of the parameters. Mescheder et al. (2017) and Nagarajan & Kolter (2017) propose to model the dynamics of the parameters using the vector field defined by its gradients. In such cases, the local convergence and stability are fully determined by the eigenvalues of the Jacobian matrix. Mescheder et al. (2018) further propose a regularization term to stabilize GANs by adjusting the eigenvalues

of the Jacobian matrix. Gidel et al. (2018) analyze the effect of momentum and propose to stabilize GANs using negative momentum. In these methods, the Jacobian matrix is the key to understand and stabilize the dynamics, whose calculation is, however, computationally expensive. It largely impedes us to generalize this analysis to complex models such as neural networks (LeCun et al., 2015).

In this paper, we argue that modeling the dynamics in the function space is more convenient for convergence analysis. Specifically, we directly model $G$ and $D$ as two dynamics whose output is considered as a signal of time $t$, i.e., representing the output of $G$ and $D$ as $G(t, z)$ and $D(t, x)$ respectively. The control theory (Kailath, 1980) provides a powerful tool to understand them and analyze their stability. Specifically, differential equations can be used to denote the dynamics, e.g., the dynamics of $G$ can be denoted as $\frac{dG(t,z)}{dt} = f(G, D)$, which represents how $G$ changes given current $G$ and $D$. By applying Laplacian transformation (Widder, 2015) to the differential equations, the dynamics can be represented as a transfer function, and then the stability and convergence can be easily modeled (Kailath, 1980) as introduced in Sec. 2.

Under the above perspective of control theory, we unify the dynamics of $G$ and $D$ as certain dynamics that are well-studied in control theory for various GANs, including Standard GAN (SGAN), WGAN and LSGAN (Mao et al., 2017), as illustrated in Sec. 3 and Appendix A. Specifically, the training process of GANs can be considered as a dynamic whose output is determined by its intrinsic properties (i.e., the objective function of $G$ and $D$) and its input (i.e., the data distribution). Within the proposed framework, a variety of existing methods (Mao et al., 2017; Mescheder et al., 2018; Gidel et al., 2018) can be considered as certain controller which is widely used in control theory as discussed in Sec. 4.1 and Appendix C. It is worth noting that through control theory, the stability of GANs can be easily inferred from the transfer function, instead of analyzing the complicated Jacobian matrix of the dynamics. The proposed framework provides a promising direction that can further benefit the training dynamics of GANs using advanced control methods. To verify it, we propose to use the widely adopted controlling method, the negative feedback (NF) control (Åström & Hägglund, 1995), to stabilize GANs. What's more, when applying NF to stable models, the performance can still be improved. The proposed NF acts as a regularization term which penalizes the $L2$ norm of the output of $D$ as we described in Sec. 4. NF is verified on the toy data such as Dirac GAN as well as the natural images such as CIFAR10 (Krizhevsky et al., 2009). The results demonstrate that our method can successfully stabilize the dynamics of GANs and outperform the baseline significantly.

## 2 PRELIMINARY

In this section, we provide the background and preliminary about Laplacian transformation (LT) and control theory. LT is a powerful tool to present an ordinary differential equation (ODE) as a rational fraction. Since most dynamics can be represented as an ODE, LT can largely simplify the analysis of stability by merely analyzing the properties of the rational fraction.

### 2.1 LAPLACIAN TRANSFORMATION

Laplacian transformation (LT) (Widder, 2015) can be considered as an extension of Fourier transformation (FT), which transforms a signal, i.e., a function of time, to a function of complex variables. Formally, the transformation and its inversion are given by:

$$\mathcal{F}(f)(s) = \int_0^\infty f(t)e^{-st}dt, \quad \mathcal{F}^{-1}(F)(t) = \frac{1}{2\pi i} \int_{c-\infty i}^{c+\infty i} e^{st}F(s)ds, \tag{1}$$

where $f$ is a signal that is represented as a function over time $t$, $c$ is a constant that ensures $F(s) \leq \infty$. $s \in \mathbb{C}$ is a complex number, i.e., $s = \sigma + \omega i$ with real numbers $\sigma$ and $\omega$ and for each $s \in \mathbb{C}$, $\overline{F}(s)$ denotes the components of certain frequency in the original signal $f$ similar to FT.

With LT, the integration and derivation can be presented as a single operator, which is given by:

$$\mathcal{F}(\frac{df(t)}{dt}) = s\mathcal{F}(f), \quad \mathcal{F}(\int_0^t f(u)du) = \frac{1}{s}\mathcal{F}(f). \tag{2}$$

With this property, an ODE can be converted to an algebraic equation by substituting derivation as an operator $s$. A simple example is illustrated as follows:

$$\frac{d^2x}{dt^2} = -x + u \rightarrow s^2X = -X + U, \tag{3}$$

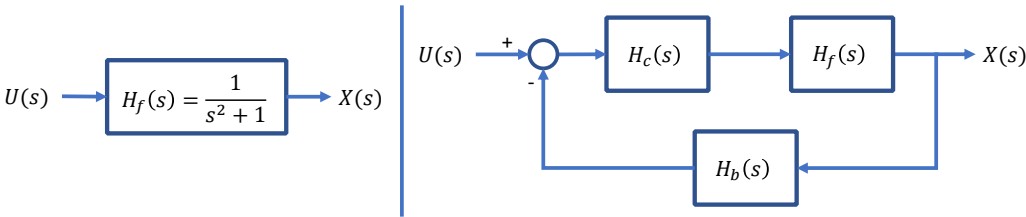

Figure 1: The diagram representations of the inputs and dynamics after LT. Left: the original diagram of the spring dynamic in Eqn. (3). Right: the diagram of a classical controlled dynamic with feedback $H_b$ and controller $H_c$.

where $u$ and $x$ are the input and output signal correspondingly and $U$ and $X$ are their Laplacian transformation. This ODE can represent a spring dynamic, whose acceleration $\frac{d^2 x}{dt^2}$ depends on the input $u$ and the current position $x$ according to Hooke's law. In this case, the ODE can be solved by simply applying the inverse Laplacian transformation to $X = \frac{1}{s^2+1}U = H_f(s)U(s)$. Here $H_f(s) = \frac{1}{s^2+1}$ is the transfer function of this dynamic which can also be represented as a diagram illustrated in Fig. 1. Since the input signal can be considered as summation of infinity Dirac signal at different positions, i.e., $f(t) = \int f(u)\delta(t-u)du$ where $\delta$ denotes the Dirac delta function (Dirac, 1981), we only need to analyze the output of the dynamics for Dirac signal whose LT is constant, i.e., $\mathcal{F}(\delta) = 1$. Therefore, the transfer function represents the intrinsic properties of the dynamics.

## 2.2 CONTROL THEORY AND NEGATIVE FEEDBACK

Given a dynamic, or equivalently a transfer function, the stability can be analyzed according to the roots of the denominator of the transfer function, e.g., $s^2 + 1 = 0$ for Eqn. (3). The roots are called the poles of the dynamics. Specifically, if all poles' real parts are negative, the dynamic is stable; if there are poles which are pure imaginary numbers, the dynamic will oscillate; if there is at least one pole with positive real parts, the dynamic will diverge (Kailath, 1980). In the above example, the poles of the dynamic are $0 \pm i$, which indicates that the object will oscillate around the original.

For those unstable dynamics, we need to stabilize it by adjusting the poles of the dynamics. One of the most commonly adopted methods here is introducing negative feedback with an auxiliary controller (Kailath, 1980). The diagram is illustrated in Fig. 1 (right). The $H_b(s)$ denotes the transfer function of feedback and $H_c(s)$ denotes the transfer function of controller.

In this setting, the dynamic takes the error (i.e., $E(s) = U(s) - H_b(s)X(s)$) rather than $U(s)$ as input, resulting the fact that the transfer function of the whole controlled dynamic is given by:

$$E(s) = U(s) - H_b(s)X(s), X(s) = H_c(s)H_f(s)E(s) \to X = \frac{H_c(s)H_f(s)}{1 + H_c(s)H_f(s)H_b(s)}U, \quad (4)$$

which provides us an approach to adjust the poles of an unstable dynamic. For most cases, simply adopting the negative feedback, i.e., let $H_b = H_c = 1$, is enough to stabilize a dynamic. In the following, we provide the dynamics of GANs in the perspective of control theory and introduce the negative feedback control to stabilize the training process of GANs.

## 3 UNIFYING THE DYNAMICS OF GANs IN CONTROL THEORY

In this section, we provide a novel perspective on the dynamic of GANs. Unlike previous methods that mainly focus on the dynamics of the parameters, we directly model the output of two dynamics $G$ and $D$ in the function space. In this part, we mainly focus on the Dirac GAN as in Mescheder et al. (2018) and unregularized WGAN. This method can be extended to other GANs easily, as we illustrated in Appendix A.

To model the dynamics of GANs, we first denote the output of $G$ and $D$ as a function with respect to time $t$, which is also known as a signal, i.e., $D = D(t, x)$ and $G = G(t, z)$. Then the training process can be represented as a dynamic using an ODE, and the equilibrium is equivalent to the final value of the dynamics (Oppenheim et al., 1998), i.e., the output of the dynamics with $t \to \infty$.

### 3.1 THE DYNAMICS OF DIRAC GAN

In Dirac GAN, the data distribution is defined as $p_D(x) = \delta_c(x) = \delta(x - c)$ which is the density of an idealized point mass at point $c$. The generator distribution is defined as $p_G(x) = \delta_\theta(x)$, where $\theta$ is the learnable parameter of $G$. In this case, we use $G(t) = G(t, z)$ for simplicity as its output does

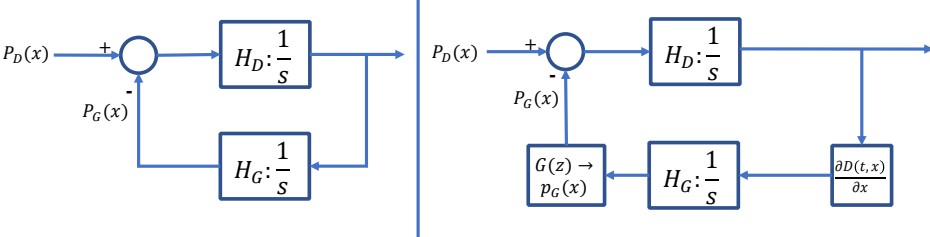

Figure 2: The diagram block of Dirac GAN and unregularized WGAN. Left: Dirac GAN, Right: unregularized WGAN.

not depend on $z$. The discriminator is a simple linear function $D(t, x) = \phi(t) \times x$. Therefore, the minimax optimization problem of Dirac GAN is formulated as:

$$\min_G \max_D \mathbb{E}_{p_D}[\phi x] - \mathbb{E}_{p_G}[\phi x] = c\phi - \theta\phi. \tag{5}$$

For the discriminator, the objective function is given by: $\max_\phi U(D) = c\phi - \theta\phi$, which indicates that the dynamics of $D$ can be formulated as:

$$\frac{\partial U(D)}{\partial \phi} = c - \theta \rightarrow \frac{d\phi(t)}{dt} = c - \theta \rightarrow \frac{\partial D(t, x)}{\partial t} = x \times (c - \theta), \forall x. \tag{6}$$

By considering $c - \theta$ as the error signal, i.e., $e = c - \theta$, the dynamics of the discriminator can be modeled as an integral part $D(t, x) = x \int_0^t c - G(t)dt$. By applying LT to the dynamic, we have:

$$\mathcal{F}(D(t, x)) = \frac{x\mathcal{F}(e)}{s} = \frac{x}{s}E(s), \forall x, \tag{7}$$

where we consider the dynamics of $D$ at each $x$. For the generator, the objective function is given by $\max_G U(G) = \theta\phi$. Therefore the dynamics of G can be formulated as:

$$\frac{\partial U(G)}{\partial \theta} = \phi \rightarrow \frac{d\theta(t)}{dt} = \phi \rightarrow \frac{\partial G(t)}{\partial t} = \frac{\partial D(t, x)}{\partial x}. \tag{8}$$

Since $D(t, x) = x \int_0^t c - G(t)dt$, we have $\frac{\partial D(t, x)}{\partial x} = \int_0^t c - G(t)dt$. We therefore combine the $x$ part in $D$'s dynamic and the the partial derivation in $G$ into an identity mapping. Then $D$ can be considered as an integral dynamic and its transfer function is given by $H_D(s) = \frac{1}{s}$. Further, the generator also integrate the output of $D$, which is also $\frac{1}{s}$.

Hence, the block diagram of the Dirac GAN is illustrated in the left panel of Fig. 2. As we can see, $D$ takes the difference between the real sample and fake sample, i.e., $c - \theta$ as input, and output the integration of the input signal. The generator takes $D$'s output as input and output the generator distribution which is the integration of $D$'s output.

**Remark 1.** When treating $G$ and $D$ as dynamics, the both the inputs and outputs of the dynamics are functions over $t$. For example, the input of $D$ is $p_D(t, x) - p_G(t, x)$ and the output is $D(t, x)$.

## 3.2 THE DYNAMICS OF WGAN

In this subsection, we further model the dynamics of WGAN (Arjovsky et al., 2017) without Lipschitz continuity constraints in the function space. In this case, we consider the nonparametric setting, where $D$ is a piece-wise linear function, $G$ is a mapping from the noise space to the data space, and both of them are of infinite capacity. This analysis can be extended to other variants of GANs easily.

In the general case, the objective function of $D$ is given by:

$$\max_D U(D) = \mathbb{E}_{p_D}[D(x)] - \mathbb{E}_{p_G}[D(x)] = \int (p_D(x) - p_G(x))D(x)dx. \tag{9}$$

According to the calculus of variations (Gelfand et al., 2000), the gradient of $U(D)$ with respect to the function $D$, and the dynamic of $D$ are given by:

$$\frac{\partial U(D)}{\partial D} = p_D - p_G \rightarrow \frac{dD(t, x)}{dt} = p_D(x) - p_G(x). \tag{10}$$

Since the input signal is $p_D - p_G$, the dynamics of WGAN is similar to that of the Dirac GAN, whose transfer function can be denoted as $H_D(s) = \frac{1}{s}$.

For the generator, the objective function and its dynamics are given by:

$$\max_{G} U(G) = \mathbb{E}_{p_z(z)}[D(G(z))] = \int p_z(z)D(G(z))dz \rightarrow \frac{\partial U(G)}{\partial G} = p_z(z)\frac{\partial D(G(z))}{\partial G(z)}, \forall z. \quad (11)$$

It is worth noting that compared to Dirac GAN, this integral is taken in the function space, i.e., $G(t + \delta_t, z) = G(t, z) + \delta_t p_z(z)\frac{\partial D(G(z))}{\partial G(z)}$. Besides, the generator distribution $p_G$ is not a Dirac distribution, resulting in that another operation, which converts the generated samples to the distribution, should be added after $G$ to make the model consistent. The diagram blocks of WGAN is illustrated in Fig. 2 (right). Compared to Dirac-GAN, another two operations are introduced to the framework. We further provide the dynamics of other GANs, including standard GAN proposed by Goodfellow et al. (2014) and Least-Square GAN proposed by Mao et al. (2017) in Appendix A.

**Remark 2.** We only consider the unregularized WGAN to simplify the analysis. Note that the dynamics of $D$ follows Eqn. (10) at least locally around the equilibrium where updating $D$ according to Eqn. (10) will not violate the Lipschitz constraints. Formally proof is given in Appendix D.

**Remark 3.** Here we model the dynamics of $G$ and $D$ in the functional space, which is more convenient for the following stability analysis and designing controllers. In practice, we update the models in the parameter space to approximate the dynamics in Eqn. (10) and Eqn. (11).

## 4 UNDERSTANDING AND STABILIZING GANS WITH NEGATIVE FEEDBACK

In this section, we first provide the stability analysis of GANs from the perspective of control theory. Besides, many previous methods can be interpreted in this perspective, and we use the momentum as an example. Other further examples are given in Appendix C. We further propose to stabilize the dynamics of GANs using the negative feedback controlling methods, and an example of another controller is given in Appendix B.

### 4.1 STABILITY

In the dynamics of GANs, we argue that it is dominated by the two integral dynamics of $G$ and $D$. Therefore, we ignore the two non-linear operations in GANs in the following analysis. Since both G and D can be considered as integral parts, we can formulate the transfer function of $G$ and $D$ as $H_G = H_D = \frac{1}{s}$. According to Eqn. (4), the transfer function of WGAN is given by:

$$H(s) = \frac{H_D}{1 + H_D H_G} = \frac{1/s}{1 + 1/s^2} = \frac{s}{1 + s^2}. \quad (12)$$

It is worth noting that the poles of the transfer function of the whole dynamic are $0 \pm i$, which indicates that the output will oscillate around the equilibrium rather than converge to 0. This result is also consistent with the analysis in (Mescheder et al., 2018). This perspective can also partially explain why larger learning rates or adopting multiple update steps for $D$, where the transfer function of $D$ can be approximated by $\frac{k}{s}$ with $k > 1$, cannot provide a principle solution for stabilizing GANs. In this case, the poles of the dynamics are adjusted to $0 \pm \sqrt{k}i$ which is still an oscillation dynamics.

### 4.1.1 THE AFFECT OF MOMENTUM

Our method can also provide stability analysis for other practical techniques used in training GANs. In this part, we use the momentum as an example. Gidel et al. (2018) provide some theoretical analysis of momentum in training GANs, which is formulated in the parameter space. In the following, we re-analyze the momentum in the function space under the perspective of control theory.

The momentum method (Qian, 1999) achieves great success in training deep neural networks. Previous methods such as SGD with momentum can help neural networks to escape from local optima. Its theoretical formulation is given by:

$$\tilde{\theta}_{t+1} = \beta\tilde{\theta}_t + (1 - \beta)\nabla\theta_t, \ \theta_{t+1} = \theta_t + \eta\tilde{\theta}_{t+1}. \quad (13)$$

The $\beta$ is the coefficient for the exponential decay of the gradients, which generally takes $0.9$. However, things changed when it comes to training GANs. DCGAN (Radford et al., 2015) recommends a smaller momentum with $\beta = 0.5$. Furthermore, recent state-of-the-art models (Mescheder et al., 2018; Brock et al., 2018; Gulrajani et al., 2017) just remove the momentum.

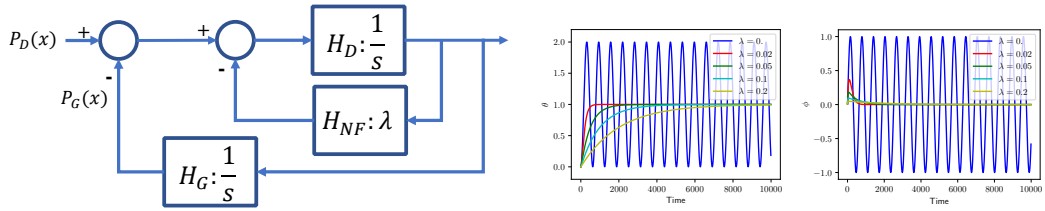

(a) The diagram of GANs with negative feedback.    (b) The dynamics of GANs with negative feedback.

Figure 3: The diagram and its corresponding dynamics of GANs using negative feedback.

---

**Algorithm 1** Negative Feedback GAN

---

1: **Input:** Buffer size $N_b$, feedback coefficient $\lambda$, batch size $N$, initialized $\theta$ and $\phi$, learning rate $\eta$.
2: Initialize $B_r$ and $B_f$ for real samples and fake samples respectively.
3: **repeat**
4:  Sample a batch of $\{x_r\} \sim p_D$, $\{x_f\} \sim p_G$ of $N$ samples.
5:  Update $B_r$ with $\{x_r\}$. Update $B_f$ with $\{x_f\}$.
6:  Sample a batch of $x'_r \sim B_r$, $x'_f \sim B_f$ of $N$ samples respectively.
7:  Estimate the objective of D:
    $U(D) = \frac{1}{N}[\sum_{x \in \{x_r\}} D(x) - \sum_{x \in \{x_f\}} D(x)] - \frac{\lambda}{2N}[\sum_{x \in \{x'_r\}} D^2(x) + \sum_{x \in \{x'_f\}} D^2(x)]$.
8:  Update $\phi$ to maximize $U(D)$ with learning rate $\eta$.
9:  Estimate the objective of $G$: $U(G) = \frac{1}{N}\sum_{x \in \{x_f\}} D(x)$.
10:  Update $\theta$ to maximize $U(G)$ with learning rate $\eta$.
11: **until** Convergence

---

In control theory, the momentum is equivalent to adding an exponential decay to the input of the dynamics (An et al., 2018), and the LT of an exponential decay dynamic is $\frac{1}{s+\tau}$. $\tau > 0$ denotes the decay coefficient which depends on $\beta$. Therefore, the transfer function of the $D$ with momentum is $H_{mD}(s) = \frac{1}{s(s+\tau)}$, and the transfer function of the WGAN with momentum is given by:

$$H(s) = \frac{H_{mD}}{1 + H_{mD}H_G} = \frac{1/(s(s+\tau))}{1 + 1/(s^2(s+\tau))} = \frac{s}{s^3 + \tau s^2 + 1}. \tag{14}$$

By letting $(s+a)(s+b)(s+c) = s^3 + \tau s^2 + 1$, we have that $a + b + c = \tau > 0$, which indicates that there are at least one pole of this dynamic whose real part is larger than 0, indicating the instability of the dynamics for GANs with momentum.

### 4.2 STABILIZE GANs

According to the above analysis, for both the unregularized WGAN and the GAN with momentum, the equilibrium point (i.e., $p_G = p_D$) is not a stable point. In the following, we propose to stabilize GANs using negative feedback control, which is the most widely-used method in control theory.

In this setting, we set $H_{NF} = \lambda$ which is the transfer function of the negative feedback for $D$. After applying the negative feedback to $D$, The diagram of the controlled dynamics is given in Fig. 3 (a). According to Eqn. (4), the transfer function of $D$ and the whole dynamic can be represented as:

$$H'_D(s) = \frac{1/s}{1 + \lambda/s} = \frac{1}{s + \lambda} \rightarrow H(s) = \frac{H'_D(s)}{1 + H'_D(s)H_G(s)} = \frac{s}{s^2 + \lambda s + 1}. \tag{15}$$

Therefore, for a positive and small $\lambda$, the real part of the two poles are $-\frac{\lambda}{2} < 0$, which indicates that the dynamic is stable. The dynamics of the Dirac GAN using different $\lambda$ is illustrated in Fig. 3 (b). As we can see, when no feedback is applied (i.e., $\lambda = 0$), the dynamic will oscillate around the equilibrium. Applying negative feedback can significant improve the stability of the dynamics.

#### 4.2.1 PRACTICAL IMPLEMENTATION

The most direct way to add negative feed is to introduce a regularization term $R_D = \frac{\lambda}{2} \int D^2(x)dx$ to the objective function of $D$. However, directly integrating over the input space is computationally infeasible. Instead, we give an approximation to $R_D$. We assume that the area where $|D(x)|$ is large mainly concentrates on its training samples, i.e., previous samples used to train $D$. Therefore, we keep two buffers $B_r$ and $B_f$ of size $N_b$ to store the old real samples and fake samples, respectively. Then $R_D$ is evaluated on these two buffers to regularize $D$, and the two buffers are updated with replacement at each step. The training procedure is illustrated in Alg. 1. In the following, we use NF-GAN $(\cdot)$ to denote our proposed method with the hyperparameters $\lambda/2$ denoted in the parentheses.

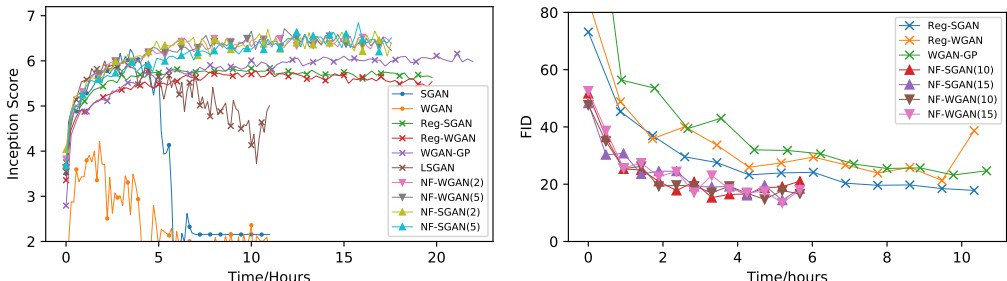

Figure 4: The learning curve of baselines and our proposed methods. Left: The Inception Score of CIFAR10. Right: The FID score for CelebA. We plot the curve with respect to time for better representation of the computational cost and convergence speed.

## 5 RELATED WORK

GANs are known for its unstable training process. Radford et al. (2015) provide several practical techniques to stabilize the training, such as designing suitable architecture of neural networks and the hyperparameters selection. The Unrolled GAN (Metz et al., 2016) proposes to make $D$ as a function of $G$ by unrolling the training process of the discriminator where the gap between the theoretical analysis and implementation can be reduced. Arjovsky et al. (2017) provide a theoretical analysis of the gradient vanishing problem and further propose to optimize GANs using Wasserstein distance. Gulrajani et al. (2017) use the gradient penalty to ensure the Lipschitz continuity in discriminator, which is required by WGAN. However, GANs still suffer from instability in the training process.

Besides, there is some work that directly focuses on the stability of the training process. Except for the works mentioned in Sec. 1, Peng et al. (2019) also directly model the dynamics of the parameters and adjust the update rules to stabilize GANs. Gidel et al. (2016) propose to find the saddle point of the optimization problem using Frank-Wolfe algorithms. Feizi et al. (2017) also use the control theory to model the stability of GANs, while the dynamics of GANs are not explicitly captured. Balduzzi et al. (2018) decompose dynamics into the oscillating and non-oscillating behaviour and model training dynamics as finding stable fixed points. However, none of them only can generalize their methods to complex datasets such as natural images.

## 6 EXPERIMENTS

In this section, we empirically verify our proposed method on the widely-adopted CIFAR10 (Krizhevsky et al., 2009) and CelebA (Liu et al., 2015) datasets. We denote the regularization proposed in Mescheder et al. (2018) as Reg-GAN, which is the direct baseline of our method. Besides, we also apply NF-GAN to SN-GAN (Miyato et al., 2018) and provide a significant improvement to the state-of-the-art method. For fair comparisons, our implementation mainly follows the officially released code for Reg-GAN and SN-GAN, respectively. More details about the experimental setting and further results on a synthetic dataset can be found in Appendix B. The code is provided here.

### 6.1 COMPARISON TO REG-GAN

We first compare our proposed method with Reg-GAN. Since for large datasets, the non-linearity cannot be ignored, which results in the gap between the simulation of Dirac GAN and our methods. A small $\lambda/2$, such as $0.01$ which is suitable for Dirac GAN is not enough for stabilize the large models. We therefore select the coefficient $\lambda/2$ among $\{1, 2, 5, 10, 15, 20\}$. In this part, we only report representative results. Though our analysis mainly focuses on the WGAN framework, NF can also easily generalize to SGAN whose stability can also be improved and NF-SGANs with different coefficient $\lambda/2$ are evaluated.

Table 1: The FID Score (top) and Inception Score (bottom) on CIFAR10. The results reported here are the best results over the training process. The results of IS are averages over 3 runs.

| Regularization | WGAN | SGAN |
|---|---|---|
| No regularization | 105.21 | 28.51 |
| Reg-GAN | 30.43 | 28.39 |
| Gradient Penalty | 28.20 | − |
| NF-GAN(2) (ours) | 21.90 | 22.35 |
| NF-GAN(5) (ours) | 22.31 | **21.61** |
| NF-GAN(10) (ours) | **21.19** | 23.34 |

| Objective | SN-GAN | NF-SN-GAN |
|---|---|---|
| WGAN | 3.29 | **8.28** $\pm$ .09 |
| Hinge | 8.22 $\pm$ .05 | **8.45** $\pm$ .11 |

We use the Inception Score (IS) (Salimans et al., 2016) to evaluate the image quality on CIFAR10 and FID score (Gulrajani et al., 2017) on both CIFAR10 and CelebA. The quantitative results are given in Table 1 and the learning curve is reported in Fig. 4. Our method gives an improvement to the FID

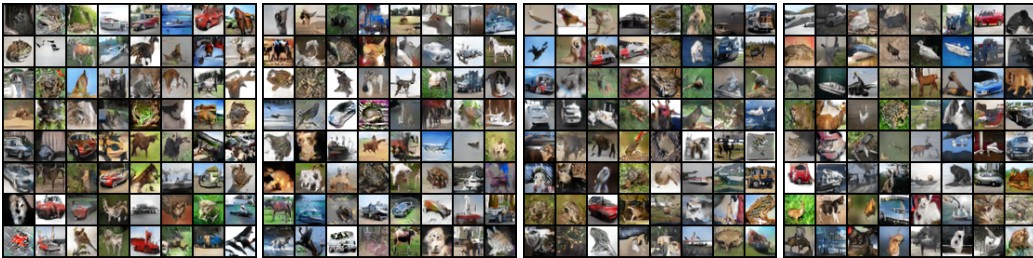

Figure 5: The generated results of CIFAR10 dataset. From left to right: WGAN-GP, Reg-WGAN, NF-WGAN(5), NF-SGAN(5).

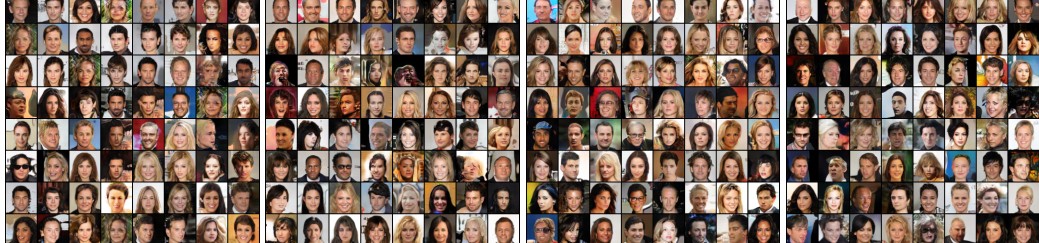

Figure 6: The generated results of CelebA dataset. From left to right: WGAN-GP, Reg-WGAN, NF-WGAN(15), NF-SGAN(15).

and IS for both WGAN and SGAN. Specifically, for unregularized methods, the models demonstrate the unstable training process and finally diverge away from the data distribution. Compared with Reg-GAN, our method maintains stability and provides better convergence results, as also shown in Table 1 (top). Besides, our method is more computationally efficient, since regularizing the output of $D$ needs less computation compared to regularizing $||\frac{\partial D(x)}{\partial x}||$ and this gap is significant for large dataset. Specifically, on Geforce 1080Ti, our method can conduct approximate $12/8$ iterations per second of training for CIFAR10/CelebA, whereas Reg-GAN can only conduct $10/4$ iterations per second. What's more, our method is robust to the choice of the hyperparameter $\lambda$. We present the generated images in Fig. 5 and Fig. 6.

### 6.2 COMPARISON TO SN-GAN

We further apply NF-GAN to the SN-GAN's architecture. With stability guaranteed, our proposed method can further help SN-GAN to address the potentially unstable issue and converge to better final results, as illustrated in Table 1 (bottom). We manually select $\lambda/2 \in \{0.01, 0.1, 1\}$ and finally adopt $\lambda/2 = 0.1$. Though SN-GAN provides an elegant normalization method to ensure the Lipschitz continuity, SN-GAN with WGAN loss still cannot converge to the data distribution and provide unreasonable results, which verifies with our stability analysis. Instead, the hinge loss, as introduced Eqn. (16)&(17) in Miyato et al. (2018), is used to stabilize SN-GAN but no theoretical guarantee is provided. In contrast, NF-GAN can successfully regularize SN-GAN with WGAN loss and provide better results to previous state-of-the-art results. What's more, NF-GAN can also benefit SN-GAN with Hinge loss and provide a significant improvement the IS of CIFAR-10.

## 7 CONCLUSION

In this paper, we propose a novel perspective to analyze the dynamics of GANs. Instead of focusing on the equilibrium, we directly model the dynamics of the discriminator and the generator during the training process. Using Laplacian transformation and control theory, the stability can be easily understood according to the poles of the dynamics. Various methods can be motivated from this perspective, and we verify it with the widely used negative feedback control. The empirical results demonstrate that our method can successfully stabilize GANs and provide better convergence results.

Negative feedback shows promise results, but further analysis is required for better results. For one thing, our analysis mainly focuses on the continuous cases, where the practical implementation optimizes both $G$ and $D$ in discrete time steps, where the $Z$-transformation provides better tools for discrete-time dynamics. For another, two approximations are made: using the update in the parameter space to approximate the dynamics and ignore two non-linear operations in GANs. Recent analyses of GANs on the functional spaces (Johnson & Zhang, 2018) provides a promising solution to solve the first approximation and modern control theory and non-linear control methods (Khalil, 2002) can be adopted to solve the second one. Indeed, our method does not provide a perfect solution to GANs' dynamics. It does provide a novel perspective to understand and stabilize GANs.

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

# A  GANS AND THEIR CORRESPONDING DYNAMICS

In this section, we provide an illustration for different kinds of GANs and their corresponding dynamics.

## A.1  VANILLA GAN AND NON-SATURATION GAN

The GANs proposed by (Goodfellow et al., 2014) are formulated as follows:

$$\min_G \max_D \mathbb{E}_{p_D}[\log \sigma(D(x))] + \mathbb{E}_{p_G}[\log(1 - \sigma(D(x)))].$$

Therefore, the objective function and the corresponding dynamics of $D$ are given by:

- The objective function:

$$
\max_D \; U(D) = \mathbb{E}_{p_D}[\log \sigma(D(x))] + \mathbb{E}_{p_G}[\log(1 - \sigma(D(x)))]
$$

$$
= \int p_D(x) \log \frac{1}{1 + \exp(-D(x))} + p_G(x) \log \frac{\exp(-D(x))}{1+\exp(-D(x))} dx
$$

$$
= - \int p_D(x) \log(1 + \exp(-D(x))) + p_G(x) \log(1 + \exp(D(x))) dx.
$$

- Given current $G$, the gradient of $U(D)$ with respect to $D$ at certain $x$ is given by:

$$
\frac{\partial U(D)}{dD(x)} = p_D \frac{\exp(-D(x))}{1 + \exp(-D(x))} - p_G \frac{\exp(D(x))}{1 + \exp(D(x))}
$$

$$
= p_D(1 - \sigma(D(x))) - p_G \sigma(D(x))
$$

$$
\rightarrow \frac{dD_t(x)}{dt} = \frac{\partial U(D(x))}{dD(x)} = p_D(1 - \sigma(D(x))) - p_G \sigma(D(x)).
$$

The objective function and the corresponding dynamics of $G$ are given by:

- The objective function:

$$
\min_G \; U(G) = \mathbb{E}_{p_G}[\log(1 - \sigma(D(x)))] = \mathbb{E}_{p_z}[\log(1 - \sigma(D(G(z))))]
$$

$$
= \int p_z(z) \log \frac{\exp(-D(G(z)))}{1 + \exp(-D(G(z)))} dz
$$

$$
= -1 \times \int p_z(z) \log(1 + \exp(D(G(z)))) dz.
$$

- Since $G$ is optimized using reparameterization, we rewrite $G(z)$ as $x_z$ for simplicity. Given current $D$, the gradient of $U(G)$ w.r.t. $G$ at a certain $z$ is given by:

$$
\frac{\partial U(G)}{\partial x_z} = -1 \times p(z) \frac{\exp(D(x_z))}{1 + \exp(D(x_z))} \frac{\partial D(x_z)}{\partial x_z} = -1 \times p(z) \sigma(D(x_z)) \frac{\partial D(x_z)}{\partial x_z}
$$

$$
\rightarrow \frac{dG_t(z)}{dt} = p(z) \sigma(D(x_z)) \frac{\partial D(x_z)}{\partial x_z}
$$

For a well-trained $D$, which can successfully tell real samples from generated samples with high confidence, i.e., $D(x_z) \approx 0$ for all $z$, the gradient for G will tend to zero almost everywhere, which will result in gradient vanishing. This analysis is consistent with previous literature (Johnson & Zhang, 2018; Arjovsky et al., 2017).

For non-saturation GAN, the objective function and dynamics of $G$ are given by:

- The objective is given by:

$$
\max_G \; U(G) = \mathbb{E}_{p_G}[\log \sigma(D(x))] = \mathbb{E}_{p_z}[\log \sigma(D(G(z)))]
$$

$$
= \int p_z(z) \log \frac{\exp(D(G(z)))}{1 + \exp(D(G(z)))} dz
$$

$$
= \int p_z(z)(D(G(z)) - \log(1 + \exp(D(G(z)))))
$$

- We rewrite $G(z)$ as $x_z$ and provide the gradient of $U(G)$ w.r.t. $x_z$ as follows:

$$\frac{\partial U(G)}{\partial x_z} = p_z(z)(1 - \frac{\exp(D(x_z))}{1 + \exp(D(x_z))})\frac{\partial D(x_z)}{\partial x_z}$$

$$= p_z(z)(1 - \sigma(D(x_z)))\frac{\partial D(x_z)}{\partial x_z}$$

$$\to \frac{dG_t(z)}{dt} = p(z)(1 - \sigma(D(x_z)))\frac{\partial D(x_z)}{\partial x_z}$$

## A.2  LEAST-SQUARE GAN

(Mao et al., 2017) propose to use the least square loss to train both the discriminator and the generator. The objective for the $D$ and $G$ are selected differently according to its theoretical analysis. Specifically, the objective function of $D$ and $G$ are given by:

$$\min_D U(D) = \frac{1}{2}\mathbb{E}_{p_D}[(D(x) - 1)^2] + \frac{1}{2}\mathbb{E}_{p_G}[(D(x))^2];$$

$$\min_G U(G) = \frac{1}{2}\mathbb{E}_{p_z}[(D(G(z)) - 1)^2].$$

Therefore, the dynamics of the $D$ and $G$ are given by:

- The dynamic of $D$:

$$\frac{\partial U(D)}{\partial D(x)} = \frac{1}{2}\frac{\partial}{\partial D(x)}(p_D(D(x) - 1)^2 + p_G D(x)^2)$$

$$= p_D(D(x) - 1) + p_G D(x).$$

$$\to \frac{dD_t(x)}{dt} = p_D - (p_D + p_G)D_t(x) \tag{16}$$

- The dynamic of $G$:

$$\frac{\partial U(G)}{\partial x_z} = p_z(D(x_z) - 1)\frac{\partial D(x_z)}{\partial x_z}$$

$$\to \frac{dG_t(z)}{dt} = p_z(z)(1 - D(x_z))\frac{\partial D(x_z)}{\partial x_z}$$

It is worth noting that if the $G$ is fixed, the dynamics of $D$ can be considered as an exponential decay which is the same as the $D$ with negative feedback control. However, since $G$ is changing during the training process, the dynamics of $D$ is actually coupling with the dynamics of $G$, which generally results in the unstable dynamics (Kailath, 1980). The experimental results on CIFAR10 also verify this argument. It also explains why we use two buffers of old samples to approximate the regularization term instead of directly regularizing the output of $D$ using current samples.

## B  FURTHER EXPERIMENTAL RESULTS

### B.1  DETAILED EXPERIMENTAL SETTINGS

In this paper, we mainly focus on the stability of the training dynamics of GANs, and therefore directly compare our proposed method NF-GAN with the vanilla SGAN/WGAN and the most direct baseline Reg-GAN. We use the resnet (He et al., 2016) for both the generator and the discriminator following Mescheder et al. (2018) and adopt the ReLU activation (Glorot et al., 2011). The batch size is 64, and the buffer size is set to 100 times of the batch size for all settings. We use rmsprop (Hinton et al., 2012) optimizer with learning rate as 0.0001 and $\alpha = 0.99$ for all models.

Note that we make the following two modifications to the original implementation of Reg-GAN, which results in a more challenging setting to the stability. First, in the original Reg-GAN, a

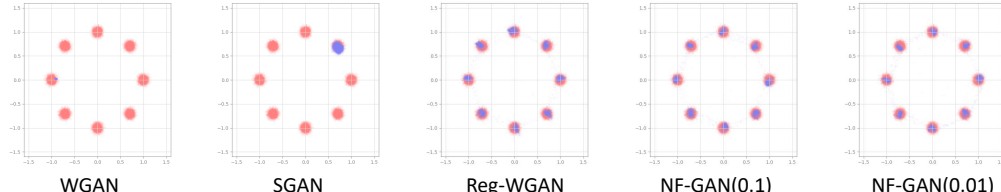

| WGAN | SGAN | Reg-WGAN | NF-GAN(0.1) | NF-GAN(0.01) |

Figure 7: The generated samples for mixture of gaussian distribution. The red points demonstrate the location of data distribution and the blue points are generated samples. Each distribution is plotted using kernel density estimation with 50,000 samples.

coefficient $0.1$ is introduced to the residual connection of the ResNet for both G and D. Specifically, $h_{L+1} = h_L + 0.1 * f_L(h_L)$ where $f_L$ is the transformation defined in the $L$-th layer and $h_L$ is the output of $L-1$-th layer as well as the input of the $L$-th layer. We instead remove the coefficient and just use the regular ResNet, i.e., $h_{L+1} = h_L + f_L(h_L)$. Second, for both G and D, we use the ReLU activation, rather than leaky ReLU. The above two modifications make the training of GANs more difficult, which explains why the reported results are not consistent with the scores in the original paper. In this more challenging setting, our method still outperforms the Reg-GAN. Specifically, the inception score of Reg-GAN is below $6.0$ in this difficult setting and about $6.5$ in the original setting. In contrast, both NF-WGAN and NF-SGAN achieve more than $6.7$ in this difficult setting.

## B.2 Synthetic Data

In this section, we evaluate our proposed method on a mixture of Gaussian on the two dimensions. The data distribution consists of 8 2D isotropic Gaussian distributions arranged in a ring, where the radius of the ring is $1$, and the deviation of each component Gaussian distribution is $0.05$. For the coefficient $\lambda/2$, we follow the theoretical analysis and set it among $\lambda/2 \in \{0.01, 0.05, 0.1\}$. We adopt two-layer MLPs for both the generator and the discriminator whose hidden units are $128$ and $512$ respectively. The batch size is is $512$ and other settings just follow the Reg-GAN.

The generated results are illustrated in Fig. 7 and we further provide the dynamics of the generator distribution in Fig. 8 in Fig. 8. As we can see, the unregularized WGAN and SGAN suffer from severe model collapse problem and cannot cover the whole data distribution. Besides, the oscillation can be observed during the training process of WGAN: the generator distribution oscillates among the modes of data distribution. Our method can successfully cover all modes compared to the WGAN and SGAN and is comparable to Reg-GAN.

## B.3 Further Control Method: PD Controller

Besides negative feedback, the most commonly used controller is the PID controller, which consists of three parts: the partial (P), integration (I) and derivation (D). Specifically, the PID controller takes the error signal as input, and output the summation of the above three parts, whose formal definition is defined as follows:

$$c(t) = K_p e(t) + K_i \int_{u=0}^{t} e(u)du + K_d \frac{de(t)}{dt} \rightarrow H_c(s) = \frac{K_p s + K_i + K_d s^2}{s}. \qquad (17)$$

$K_p$, $K_i$, $K_d$ denotes the corresponding coefficients for PID controller. $e(t)$ denotes the error signal and $c$ is the output of the controller, which is also the actual input of the dynamic. Generally, the I part can reduce the static error, and the D part can make the dynamic stable and accelerate the convergence.

The PID controller is introduced to adjust the poles of the dynamic. Since the static error, i.e., the difference between the stable point and targeted point, is zero, the integral part is not necessary, and we only employ the PD controller in the following. By letting $H_c(s) = p + ds$, where $p$ and $d$ is the corresponding coefficient for the P and D part, the transfer function of the whole dynamic is given by:

$$H(s) = \frac{H_c(s)H_D(s)}{1 + H_c(s)H_G(s)H_D(s)} = \frac{s(p+ds)}{s^2 + ds + p}. \qquad (18)$$

As we can see, given positive $p$ and $d$, the PD controller can successfully adjust the real parts poles of the dynamic to negative. In the following analysis and implementation, we fix $p = 1$ for simplicity.

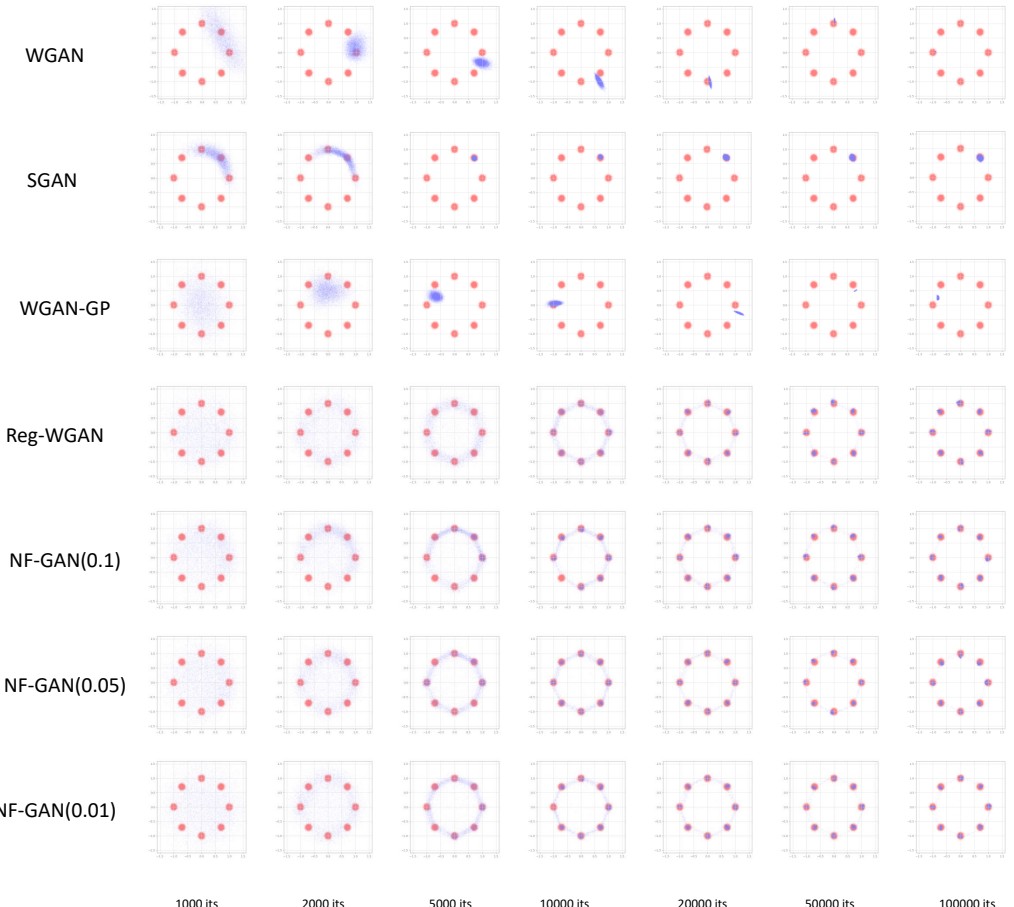

Figure 8: The training dynamics of various GANs on synthetic data.

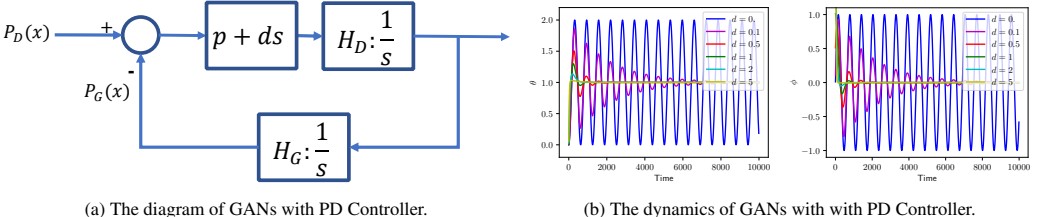

(a) The diagram of GANs with PD Controller.

(b) The dynamics of GANs with with PD Controller.

Figure 9: The diagram and its corresponding dynamics of GANs using PD controller.

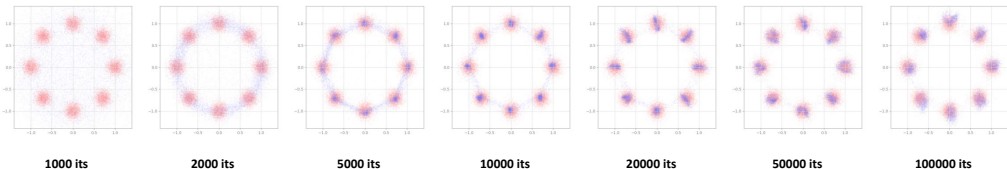

1000 its    2000 its    5000 its    10000 its    20000 its    50000 its    100000 its

Figure 10: The generated results of PD controller with $d = 5$.

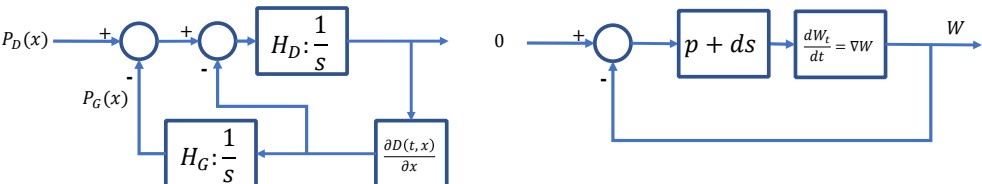

Figure 11: The diagram of previous methods. Left: the diagram of Reg-GAN which takes the $||\frac{\partial D}{\partial x}||$ as regularization. Right: the diagram of negative momentum together with weight decay.

The dynamics of Dirac GANs using PD controller is given in Fig. 9 (b). With a PD controller, the dynamic can be successfully stabilized. For general GAN, the objective of $D$ at time $t$ is given by:

$$U(D,t) = \mathbb{E}_{p_D(t,x)}[D(x)] - \mathbb{E}_{p_G(t,x)}[D(x)]+$$
$$\lambda\{\mathbb{E}_{p_D(t,x)}[D(x)] - \mathbb{E}_{p_G(t,x)}[D(x)] - \mathbb{E}_{p_D(t-\delta t,x)}[D(x)] + \mathbb{E}_{p_G(t-\delta t,x)}[D(x)]\}. \quad (19)$$

The regularization is actually minimizing the loss of $D$ in the current step as well as maximizing the loss of D for previous step.

### B.3.1 EXPERIMENTAL RESULTS

The results of the synthetic data are illustrated in Fig. 10. WGAN with PD controller can successfully cover the whole data distribution and be stable at the equilibrium.

However, we fail to generalize PD controller to the natural image dataset such as CIFAR10. We argue that the main reason is the overshooting for the dynamics of $D$, as illustrated in Fig. 9 (b), where the $D$ gives a large derivation from the equilibrium at the beginning. Because of the highly-nonlinearity in neural networks, this overshooting will result in unexpected behavior of neural networks. In contrast, for the negative feedback, the overshooting is much smaller, which enables it to generalize to the non-linear neural networks.

## C UNIFYING PREVIOUS METHODS IN CONTROL THEORY

In this section, we also provide some analysis that interprets previous methods in the perspective of control theory.

### C.1 REG-GAN

Mescheder et al. (2018) propose a regularization which is added on $D$'s gradient of $x$, i.e., $R_d(x) = ||\frac{\partial D(x)}{\partial x}||_2^2$. Therefore, it can be considered as an other form of negative feedback, which is illustrated in Fig. 11.

Compare to our analysis, Reg-GAN provides more accurate modeling of the dynamics and directly regularizes on the input of the dynamic of $G$. Therefore, Reg-GAN can also successfully stabilize the dynamic of GANs, which is consistent with our experimental results. Compared to Reg-GAN, our method is more computationally efficient.

### C.2 WEIGHT DECAY AND NEGATIVE MOMENTUM

Gidel et al. (2018) analyze the effect of momentum in the training of GANs, and we already interpret this analysis in the context of control theory. Besides, the authors also propose to use the negative

momentum to stabilize GANs, which is illustrated as follows:

$$W_{t+1} = W_t + \eta \nabla W + \lambda(W_t - W_{t-1}). \tag{20}$$

Together with weight decay, the controlled dynamics of $W$ is given:

$$W_{t+1} = W_t + \eta \nabla W + \lambda_1(W_t - W_{t-1}) - \lambda_2 W_t. \tag{21}$$

In this setting, Eqn. (21) is equivalent to the PD controller with negative feedback in the parameter space whose diagram can be illustrated in Fig. 11. The $\lambda_1(W_t - W_{t-1})$ denotes the D part and $\lambda_2 W_t$ denotes the P part.

## D  LOCAL APPROXIMATION TO THE REGULARIZED DYNAMICS

In this section, we prove that around the equilibrium, the dynamics of regularized $D$ with Lipschitz constraint is equivalent to the unregularized $D$ as in Eqn. (10). Since the updating direction of $D$ is the gradient of $D$ in the functional space, we only need to prove that updating $D$ according to Eqn. (10) will not violate the Lipschitz constraints, at least locally around the equilibrium. Here we make the following assumptions:

1. Both $p_D(x)$ and $p_G(t, x)$ are $C^1$-smooth: $\frac{dp(x)}{dx}$ exists and is continuous for $p_D$ and $p_G(t, x), \forall t$.

2. $p(x) \to 0$ and $\frac{dp(x)}{dx} \to 0$ when $x \to 0$ for $p_D$ and $p_G(t, x), \forall t$.

3. There exists an $M$ such that $|\frac{dp(x)}{dx}|_2 < M$ for $p_D$ and $p_G(t, x), \forall t$.

The above assumptions are satisfied for most probability density functions.

The distance in the function space is defined as $d(p_1, p_2) = \sup_{x \in \mathbb{R}^n} |p_1(x) - p_2(x)|$ which always exists because of the 2-nd conditions above. We define $\Omega_L = \{p(x)|p(x) \in C^1, |\frac{dp(x)}{dx}|_2 < L \, \forall x.\}$ and $B(\epsilon) = \{p(x)|p(x) \in C^1, \sup_x |p(x)| < \epsilon\}$. Then we have the follow theorem:

**Theorem 1.** *There exists $\delta > 0$, such that $\forall D(x) \in \Omega_{0.5}$, we have $D(x) + \delta(p_D(x) - p_G(x)) \in \Omega_1$.*

*Proof.* By denoting $D'(x) = D(x) + \delta(p_D(x) - p_G(x))$, We have:

$$\frac{d(D(x) + \delta(p_D(x) - p_G(x)))}{dx} = \frac{dD(x)}{dx} + \delta(\frac{p_D(x)}{dx} - \frac{p_G(x)}{dx}).$$

Therefore, we have

$$|\frac{d(D(x) + \delta(p_D(x) - p_G(x)))}{dx}|_2 \leq |\frac{dD(x)}{dx}|_2 + \delta(|\frac{p_D(x)}{dx}|_2 + |\frac{p_G(x)}{dx}|_2) |_2 \tag{22}$$

$$\leq 0.5 + \delta(M + M). \tag{23}$$

By letting $\delta = \frac{1}{4M}$, we have $|\frac{d(D')}{dx}|_2 \leq 0.75$. Therefore we have $D'(x) \in \Omega_1$. $\qquad \square$

The above theorem indicates that when $D(x)$ is sufficient close to the equilibrium, then the dynamics of $D$ still follows Eqn. (10).

## E  CONNECTION TO REGULARIZATION ON JACOBIAN MATRIX

In this paper, we mainly analyze our proposed method in the functional space, including stability analysis and controller designing. Instead, our proposed method can also be interpreted as certain regularization terms on the Jacobian matrix of the training dynamics. Below we provide a formal demonstration.

First, we denote the equilibrium of $G$ and $D$ in the functional space as $(\theta^*, \phi^*)$, where $p_G(x; \theta^*) = p_D(x)$ and $D(x; \phi^*) = 0$ for all $x$. Therefore, we have that $\phi^*$ is also a global minimum point of the regularization term $L(D) = \int D^2(x)dx$. Then we have $\frac{\partial^2 L(D)}{\partial \phi^2} \succeq 0$.

We denote $U(D, G)$ as the objective function of the minimax optimization problem in WGAN without NF regularization. Then the Jacobian matrix of the training dynamic can be denoted as:

$$J = \begin{pmatrix} \frac{\partial^2 U(D,G)}{\partial \phi^2} & \frac{\partial^2 U(D,G)}{\partial \phi \partial \theta} \\ \frac{\partial^2 U(D,G)}{\partial \theta \partial \phi} & \frac{\partial^2 U(D,G)}{\partial \theta^2} \end{pmatrix}. \tag{24}$$

Because of the linearity of the derivation, the training dynamics of the WGAN with NF regularization is denoted as:

$$J' = J - J_L = J - \begin{pmatrix} \frac{\partial^2 L(D)}{\partial \phi^2} & \mathbf{0} \\ \mathbf{0} & \mathbf{0} \end{pmatrix}, \tag{25}$$

where we abuse the $\mathbf{0}$ to denote the zero matrix with certain size to match the size of $J$. Since $\frac{\partial^2 L(D)}{\partial \phi^2} \succeq 0$, we have $-J_L \preceq 0$. Therefore, the NF regularization introduces a negative semi-definite matrix to the original Jacobian matrix, which is helpful to stabilize the training dynamics of GANs.

