# OpenReview forum: "Understanding and Stabilizing GANs' Training Dynamics with Control Theory"
_ICLR.cc/2020/Conference — Reject_

### Official Review · AnonReviewer2 · 2019-10-20
**Official Blind Review #2**

**Rating:** 6

**Review:**

This paper proposes a novel view for stabilising GANs from the perspective of control theory. This view provides new insights into GAN training and may inspire future research along this direction. This paper is overall well written, with a smooth introduction of background material that might be less familiar for machine learning researchers. There are places that need further clarification, but I think the proposed direction is promising.

Questions about the method:

- Since the proposed method starts from Laplace transform, it would be helpful to further discuss the connection between other methods that regularises the eigenvalues of the Jacobian (such as spectral-normalisation), which work in the frequency domain from a different perspective. For example, could the proposed regulariser be interpreted as imposing certain constraint on the spectrum of Jacobian?

- Does section 2.2 depend on the assumption of linear dynamics?

- Does the E in eq.7 come from eq. 4?

- Could you give some intuition for the paragraph above section 3.4, about the different form of inputs when treating D and G as dynamics? For consistency, it is perhaps better to keep the dependency of p_D and p_G on x explicit (same for eq. 10), unless this is intended?

- My main concern about the analysis is that it shows why several methods (e.g., momentum, multiple update steps) are *not* helpful for stabilising GANs, but does not tell why training with these methods, as well as others such as gradient penalty, *do converge* in practice with properly chosen hyper-parameters?

Experiments:

- About setup: the paper reports using ResNets for natural images as in Mescheder et al. (2018). However, Mescheder et al. (2018) uses DCGAN for CIFAR10, which raises further questions about the scores on this dataset:

- The baseline scores of Reg-SGAN and Reg-WGAN seem to be worse than those reported in Mescheder et al. (2018), which have inception scores above 6 according to Figure 6 of their paper. In Figure 5 of this paper, they are clearly below 6. What’s the reason for this discrepancy?


**Experience Assessment:**

I have published one or two papers in this area.

**Review Assessment: Checking Correctness Of Derivations And Theory:**

I assessed the sensibility of the derivations and theory.

**Review Assessment: Checking Correctness Of Experiments:**

I carefully checked the experiments.

**Review Assessment: Thoroughness In Paper Reading:**

I read the paper at least twice and used my best judgement in assessing the paper.

---

> ### Author Response · Authors · 2019-11-14
> **Response to Review #2**
>
> Thanks for your valuable comments. Below we address the detailed comments.
>
> Q1: Connection to the Jacobian matrix:
> A1: Thanks for the interesting suggestion. Indeed, the proposed regularizer can be interpreted as certain constraints on the Jacobian at the equilibrium point. Since at the equilibrium, $D(x)=0$ for all x, indicating that the equilibrium is a global optimal point of the negative feedback regularization $L = \lambda \int D^2(x)dx$. Therefore, the Hessian matrix $J = \frac{\partial^2 L}{\partial \phi^2}$ is positive-semidefinite. Otherwise, $\phi$ is a stationary point instead of the global optimal point of the regularization term. Therefore, introducing the $L$ is equivalent to adding a negative-semidefinite matrix to the jacobian matrix of the original dynamics, which do help to stabilize the dynamics. We added the related discussion in Appendix E in the revision.
>
> Q2: The linear assumption on the dynamics:
> A2: Yes. The Laplacian transformation and the following discussions in Sec. 2.2 rely on linear dynamics, but it does not put any restriction on defining the discriminator as a nonlinear neural network. In Section 3, we can see that in the function space, the discriminator $D(x)$ and the generated samples $G(z)$ can be considered as integral parts which are also linear dynamics. The two non-linear operations are clearly denoted in Fig. 2(right), and we make an approximation to ignore these two non-linear operations which is widely adopted in control theory [*1].
>
> Q3: Eqn. (7):
> A3: Actually, Eqn. (7) comes from Eqn. (6). Letting $e = c - \theta$, and taking Laplacian transformation on both side of Eqn. (6), we have $s\mathcal{F}(D(t, x)) = x \mathcal{F}(c-\theta)$, which induces Eqn. (7). We made this clearer in the revision.
>
> Q4: The input and output of dynamics:
> A4: For a dynamic, both the input and the output are functions of time $t$. We take $D$ as an example. Since the dynamics of $D$ is equivalent to the integral part, the output of $D$ can be formulated as $D(t, x) = \int_0^{t} g(u, x)du$, where $g$ is also a function of time t. In this setting, we say that the input of the dynamic is $g(t, x)$ and the output is $D(t, x)$, for all $x$. In Eqn. (10), we ignore the $x$ to emphasize that we modeling $D$ in the time-space. We added an example for a better presentation in Sec. 3.1 in the revision.
>
> Q5: The stability of previous methods:
> A5: Indeed, many existing methods can *generate realistic images*, which, however, does not necessarily imply that these methods *are stable*. For example, in Fig. 4 (left), the inception score of SGAN and LS-GAN is high at the beginning but finally diverges after a period. The early stopping in GAN's training is widely adopted otherwise the image quality will decrease, which indicates that these methods are not actually stable. Besides, NF-GAN can also boost the performance of SN-GAN to achieve new state-of-the-art performance (see details below). It indicates that improvement on the stability also benefits the state-of-the-art methods.
>
> Q6: Experiments:
> A6: For the experiments, we directly used the officially released code of Reg-GAN for fair comparison and it uses the ResNet instead of DCGAN architectures. We provided a detailed explanation about the experimental setting and further experimental results of the state-of-the-art performance in our response to "The Common concerns about experimental setting and results".
>
> [*1] Khalil, Hassan K. "Nonlinear systems." Upper Saddle River (2002).

---

### Official Review · AnonReviewer3 · 2019-10-23
**Official Blind Review #3**

**Rating:** 3

**Review:**

This paper gives a perspective of GANs from control theory, where it is well established in which cases a dynamical system (which can be described analytically as a function of time) is stable or not (by looking at the roots of the denominator of the so-called transfer function in control theory). It is interesting that the analysis using this framework on simple examples is in line with known results in the GAN literature (Dirac GAN).

Although I personally enjoyed reading the results that from control theory perspective are inline with GAN literature, the paper does not provide novel surprising results. For e.g. the results on the oscillating behavior of Dirac-GAN are described in related works (e.g. Mescheder et al. 2018), and in practice, WGAN with no regularization is not used (as well as GAN with momentum, as normally beta1=0 in practice). In my understanding, the authors present these results to justify the validity of the approach. However, this limits the novelty of the results relative to existing literature. The authors do not focus (in the main paper) on GAN variants used currently, and it is not clear if the proposed approach provides improvement relative to the current state of the art implementations (see next paragraph). Moreover, if I understand correctly the WGAN analysis does not take into account that G and D are non-linear, and it is unclear if these can be done.

I am also wondering if the comparison with the baselines is fair. In other words, although in the present results, the proposed NF-SGAN/WGAN outperforms the baseline, the reported performance of the baselines is worse than in related works on CIFAR10. In particular, FID of ~30 on CIFAR10 for the baselines is notably higher then current reported results on this dataset (e.g. Miyato et al. 2018; Chavdarova et al. 2019). In my opinion, the authors could start from the existing state of the art implementations on this dataset, and report if negative feedback (NF) improves upon.

As the approach uses NF is derived specifically for unstable dynamics, it is not clear to me how adding it would affect the training if the dynamics *is stable*. In this context, I consider that for example, the work by Balduzzi et al. 2018 may be relevant as it describes that the dynamics of games (the Jacobian) has two components, one of which describes the oscillating behavior (Hamiltonian game); whereas most games are a mix of oscillating and non-oscillating dynamics. It is not clear to me if NF would improve stability/performances in general games. As the authors’ main claim is improved stability I am curious to see more detailed analysis on real-world datasets (e.g. multiple seed runs, 2nd-moment estimates over iterations as in Chavdarova et al. 2019).

In summary, although the proposed perspective seems promising given the presented results and it is interesting, in my opinion, it does not provide novel insights nor obtains current state-of-the-art results on CIFAR10, or guarantee that if pursuing it would allow for solving current issues of GAN training.

--- Minor ---
- Abstract: ‘converge better’ it is not clear to me in what sense (faster/better final performances)
- Page 3, Eq. 5: as D and G are functions of time here, eq 5 should maybe be written in more detail to include t
-  Page 3, Sec. 3: I think citing works that also focus on Dirac-GAN would motivate better why you focus on Dirac-GAN in this paper (e.g. writing `as in Mescheder et al. 2018`)
- Page 4: infinity - infinite
- Page 5: can also.. explains ->  explain


**Experience Assessment:**

I have published one or two papers in this area.

**Review Assessment: Checking Correctness Of Derivations And Theory:**

I assessed the sensibility of the derivations and theory.

**Review Assessment: Checking Correctness Of Experiments:**

I assessed the sensibility of the experiments.

**Review Assessment: Thoroughness In Paper Reading:**

I read the paper at least twice and used my best judgement in assessing the paper.

---

> ### Author Response · Authors · 2019-11-14
> **Response to Reviewer #3 (1/2)**
>
> Thanks for acknowledging our novel perspective as well as giving the valuable comments. Below, we address the detailed comments. Particularly, we clarify some potential misunderstandings and provide new results to show state-of-the-art results.
>
> Q1: About the main concern on “novelty, improvement relative to the current state of the art implementations,  and non-linearity of G and D”:
> A1: As for novelty, we first thank you for acknowledging that understanding GANs from the control theory perspective is promising and enjoyable to read. Indeed, as agreed by R#2, this is a major novel contribution that provides a unified and promising framework to model the stability of GANs, which includes some recent developments (e.g., Negative Momentum and Reg-GAN, See Sec. 4.1 and Appendix A&C) and also provides us a possibility to explore advanced tools in control theory (e.g., nonlinear control and modern control theory [*3]) to improve both the stability and convergence speed of GANs. Then, as some useful examples, in this paper, we particularly showed that the technique of negative feedback can be leveraged to stabilize GANs and developed NF-GAN, which was proven to be effective in our experiments. Finally, we added new results in Table 1 in the revision, which shows that the same technique of negative feedback can further improve the state-of-the-art method of SN-GAN [*5]. Specifically, we apply NF-GAN to SN-GAN [*5] and NF-GAN provides a significant improvement on the state-of-the-art inception score on CIFAR-10 (from 8.22 to 8.45). Such results indicate that our technique of NF-GAN can still benefit the state-of-the-art variants of GANs (e.g., SN-GAN).
>
> Overall, our perspective is novel and it indeed sets new state-of-the-art results as compared to the current variants of GANs.
>
> As for the statement “the WGAN analysis does not take into account that G and D are non-linear”, this is a potential misunderstanding. In fact, in the WGAN analysis, we do not put any constraints on the G and D networks, which can be any well-defined nonlinear models. This confusion may arise from the linearity of the dynamics. In Sec. 3.2, as we model the dynamics of G and D in the functional space instead of the parameter space, they can both denoted as integral parts (See Eqn. (10)&(11)), thereby both are linear. However, this doesn’t influence the nonlinearity of G and D with respect to the weights. Technically, we provide an approximate solution to deal with such nonlinearity. As discussed in Remark 3 and Section 7 of the revised paper, we also note that the recent analyses of GANs on the functional spaces [*4] can provide a promising solution to solve this approximation and we leave it as our future work.
>
>
> Q2: About “comparison with the baselines”:
> A2: For fairness, we tried our best to fairly compare all methods. See details in the Concern 1 of our post for common concerns. For the state-of-the-art results, we indeed provided a significant improvement on the inception score of CIFAR-10 (from 8.22 to 8.45) using SN-GAN’s architecture as suggested.

---

> > ### Author Response · Authors · 2019-11-14
> > **Response to Reviewer #3 (2/2)**
> >
> >
> > Q3: How does negative feedback (NF) influence the training of stable dynamics and further evaluation:
> > A3: As stated above in our response to Q1, we added the new results of applying negative feedback to SN-GAN, which is a state-of-the-art variant of GANs with empirically stable performance. Our results (See Table 1 (bottom) in the revision) indeed show that NF can further improve to reach new state-of-the-art results. In general, as NF is essentially a penalty term that regularizes $D$ to the zero-function, we can expect it to be effective for most dynamics [*3]. We also included the suggested related work (Balduzzi et al. 2018) in Section 5. Finally, as for the further evaluation (such as multiple seed runs and 2nd-momentum estimates), we agree it is interesting, but it is very demanding in computational resources, and we leave it for a systematic future investigation.
> >
> >
> >
> > In summary, this paper proposes a unified framework to model the dynamics of GANs which is a powerful tool to stabilize and improve GANs. The experimental results on SN-GANs demonstrate that NF-GAN can further improve the performance of stable GANs and provide an improvement to the state-of-the-art models. Finally, it is a promising direction to follow where further advanced control methods can benefit the training of GANs.
> >
> > [*1] Gidel, Gauthier, et al. "Negative Momentum for Improved Game Dynamics." The 22nd International Conference on Artificial Intelligence and Statistics. 2019.
> > [*2] Mescheder, Lars, Andreas Geiger, and Sebastian Nowozin. "Which Training Methods for GANs do actually Converge?." International Conference on Machine Learning. 2018.
> > [*3] Khalil, Hassan K. "Nonlinear systems." Upper Saddle River (2002).
> > [*4] Johnson, Rie, and Tong Zhang. "Composite Functional Gradient Learning of Generative Adversarial Models." International Conference on Machine Learning. 2018.
> > [*5] Miyato, Takeru, et al. "Spectral Normalization for Generative Adversarial Networks." (2018).

---

### Official Review · AnonReviewer1 · 2019-10-23
**Official Blind Review #1**

**Rating:** 3

**Review:**

Authors use control theory to analyze and stabilize GAN's training. Their method, effectively, adds an L2 regularization to the output of the discriminator.

I have some concerns regarding the novelty, analysis and also the experiments.

- The analysis has focused on a very simple case of having a linear discriminator which for example in WGAN, forces the first moments to match. How does the analysis extend to more realistic cases?

- In eq 9 in the dynamics of WGAN section, the discriminator should be restricted to Lip functions. This has not been considered in the analysis.

- There are a few work in the literature that analyze local stability of GANs (e.g. https://arxiv.org/abs/1706.04156) as well as using some control theory for analyzing global stability of GANs (e.g. https://arxiv.org/abs/1710.10793). The connections of the proposed approach with existing literature should be better explained.

- In the empirical results, the performance of the proposed method and Reg-GAN (from the numerics of GAN paper) are quite similar. Are there instances that the proposed approach significantly improves the stability of practical GAN architectures?



**Experience Assessment:**

I have published in this field for several years.

**Review Assessment: Checking Correctness Of Derivations And Theory:**

I assessed the sensibility of the derivations and theory.

**Review Assessment: Checking Correctness Of Experiments:**

I assessed the sensibility of the experiments.

**Review Assessment: Thoroughness In Paper Reading:**

I made a quick assessment of this paper.

---

> ### Author Response · Authors · 2019-11-14
> **Response to Reviewer #1 (1/2)**
>
> Thanks for your comments. Below we address the detailed comments. In particular, we clarify the potential misunderstanding on the linearity of the discriminator and added new state-of-the-art results by applying negative feedback to SN-GAN.
>
> Q1: About novelty and analysis:
> A1: As agreed by both Reviewer #2 and Reviewer #3, in this paper, our contributions are twofold: (1) a unified and promising framework to model the stability of GANs using control theory, (2) we propose to use the negative feedback to stabilize GANs.
>
> First, using control theory, the dynamics of GANs can be modeled as transfer functions with Laplacian transformation, and various existing methods (e.g., Negative Momentum and Reg-GAN) can be considered as certain controllers that are widely used in control theory. Moreover, through control theory, the stability of GANs can be easily inferred from the poles of the transfer function, instead of analyzing the complicated jacobian matrix of the dynamics as discussed in Sec. 4.1 and Appendix A&C. We argue that our method is distinct from existing method, which is well-discussed in our response to Q4.
>
> Second, our perspective also provides a promising direction that can further benefit the training dynamics of GANs using advanced control methods (e.g., nonlinear control and  modern control theory [*2]) to improve both the stability and the convergence speed of GAN. As a concrete example, we propose to use the most widely-used negative feedback control method to stabilize GAN's dynamics and the empirical results demonstrate the effectiveness of NF-GAN as shown in Sec. 4&6. Exploring advanced control methods is our important future work. We updated the empirical results on the state-of-the-art model in the revision, where we applied our proposed NF-GAN to the SN-GAN [*6]. We can see that NF-GAN can successfully address the potential unstable issues of SN-GAN and achieve state-of-the-art inception score on CIFAR-10. More details can be found in our response to the common concern and our revised paper.
>
> Q2: Linear discriminator and extending the analysis to realistic settings:
> A2: Thanks. We indeed extended the analysis of Dirac GAN to the more realistic setting in Sec. 3.2, where the discriminator is NOT linear. In this part, we analyzed the dynamics of WGAN in the function space following [*1], i.e., we directly modeled $D(t, x)$ and $G(t, z)$ for all $x$ and $z$. It avoids the nonlinearity issue caused by the neural network, and both G and D are linear dynamics, at least locally around the equilibrium, as discussed in Sec. 3.2 and Appendix D in the revision. Fig. 2 (right) provides a diagram of the unregularized WGAN. In practice, we use the gradient descent method in the parameter space to approximate the dynamics in the functional space to efficiently solve the optimization problem. Recent advances in modeling GAN in the functional space [*5] provide powerful tools to bridge the gap and we leave it as our future work. We updated the discussion in Sec. 3.2 in the revision to make this clearer.

---

> > ### Author Response · Authors · 2019-11-14
> > **Response to Reviewer #1 (2/2)**
> >
> >
> > Q3: The Lip constraints on the discriminator:
> > A3: Actually, our method also applies to WGAN with Lipschitz constraints (vanilla WGAN). Existing work [*3] states that vanilla WGAN diverges and we provide theoretical and empirical evidence that our method helps vanilla WGAN converge.
> >
> > Theoretically, to address this comment, we added Theorem 1 (See in the Appendix D) that states the dynamics of $D$ with Lipschitz constraint follows Eqn. (10) *around the equilibrium*. Therefore, the stability analysis and our proposed method in Sec. 4 still applies to vanilla WGAN because control theory mainly focuses on the stability *around the equilibrium* [*2].
> >
> > Empirically, as suggested by R#3, we built a vanilla WGAN baseline using the SN-GAN [*6] framework, whose Lipschitz constraints are satisfied through spectral normalizations. We compared SN-GAN (WGAN loss) and NF-SN-GAN (WGAN loss) and obtained a significant improvement on both the stability and the final results (IS from 3.29 to 8.28, See details in our post for common concerns). It demonstrates that our method helps vanilla WGAN converge, which is consistent with our theoretical analysis.
> >
> > Q4: Related work:
> > A4: Thanks for pointing out the related work. In fact, our method is distinct from these methods.
> >
> > For the first paper (i.e., Gradient descent GAN optimization is locally stable) analyzed the stability of GANs using the Jacobian matrix and adopted a regularization term to stabilize GANs similarly to [*4]. Instead, we adopted a different method to model the dynamics from control theory. The difference has been discussed in Sec. 1 and Sec. 5
> >
> > For the second paper, the authors used the Lyapunov function, which is different from our framework, to analyze the stability of GANs. Besides, their method fails to scale-up to large datasets such as CIFAR-10 because of computational issues.
> >
> > Q5: Empirical results:
> > A5: Theoretically, Reg-GAN is also a stable training method for GANs but it is computationally less efficient than NF-GAN (ours), as illustrated in Fig. 4. Empirically, we can achieve better results compared to Reg-GAN as illustrated in Table 1 (top).
> >
> > Moreover, we also also advanced the state-of-the-art results based on practical GANs (SN-GAN). The inception score on CIFAR-10 is improved from 8.22 to 8.45. See details in Table 1 (bottom) in the revision and our post about common concerns.
> >
> > [*1] Johnson, Rie, and Tong Zhang. "Composite Functional Gradient Learning of Generative Adversarial Models." International Conference on Machine Learning. 2018.
> > [*2] Khalil, Hassan K. "Nonlinear systems." Upper Saddle River (2002).
> > [*3] Mescheder, Lars, Andreas Geiger, and Sebastian Nowozin. "Which Training Methods for GANs do actually Converge?." International Conference on Machine Learning. 2018.
> > [*4] Mescheder, Lars, Sebastian Nowozin, and Andreas Geiger. "The numerics of gans." Advances in Neural Information Processing Systems. 2017.
> > [*5] Johnson, Rie, and Tong Zhang. "Composite Functional Gradient Learning of Generative Adversarial Models." International Conference on Machine Learning. 2018.
> > [*6] Miyato, Takeru, et al. "Spectral Normalization for Generative Adversarial Networks." (2018).

---

### Author Response · Authors · 2019-11-14
**Response to the common concerns about experimental setting and results (R1, R2, R3)**


Concern 1: Fairness about the experimental settings.
A1: In this paper, we used *exactly the same setting* in all empirical comparisons reported in our paper for fairness. As for the Reg-GAN, we used a more challenging setting than the one in the original paper to demonstrate the effectiveness of our method, which explains why the reported results are not consistent with the scores in the original paper. We emphasize that NF-WGAN and NF-SGAN outperform the results published in the Reg-GAN paper even in the challenging setting. Below, we compare the settings and results in detail.

For the settings, we made the following two modifications to the original implementation of Reg-GAN, which results in a more challenging setting to the stability. First, in the original Reg-GAN, a coefficient $0.1$ is introduced to the residual connection of the ResNet for both G and D. Specifically, $h_{L+1} = h_{L} + 0.1*f_L(h_L)$ where $f_L$ is the transformation defined in the $L$-th layer and $h_L$ is the output of $L-1$-th layer as well as the input of the $L$-th layer. We instead removed the coefficient and just used the regular ResNet, i.e., $h_{L+1} = h_{L} + f_L(h_L)$. Second, for both G and D, we used the ReLU activation, rather than leaky ReLU. The above two modifications make the training of GANs more difficult, which explains why the reported results are not consistent with the scores in the original paper.

For the results, in this more challenging setting, our method still outperforms the Reg-GAN. Specifically, the inception score of Reg-GAN is below 6.0 in this challenging setting and about 6.5 in the original setting. In contrast, both NF-WGAN and NF-SGAN achieve more than 6.7 in this challenging setting. The above discussion is updated in Appendix B.1 in the revision.


Concern 2: State-of-the-art models
A2: We achieved state-the-art results in newly added experiments based on SN-GAN [*1], as suggested by Reviewer #3. We adopted the official implementation of SN-GAN and used exactly the same setting as the original paper. The results are reported in the following table.

Table: Inception score for both SN-GAN and NF-SNGAN. The reported results of NF-SNGAN are averaged over three runs.
-------------------------------------------------------------------------
| Objective Function | SN-GAN        | NF-SNGAN            |
| WGAN loss               | 3.29               | 8.28 $\pm$ 0.09         |
| Hinge loss                | 8.22  $\pm$ 0.05 | 8.45 $\pm$ 0.11         |
-------------------------------------------------------------------------

According to the results, our method is effective in two aspects. First, SN-GAN largely relies on the proper objective function (Hinge loss). In contrast, our method achieves outstanding results with both objective functions. It shows the promise of our method as a general framework to understand and stabilize GANs. Second, NF-GAN can advance the state-of-the-art GANs on the inception score of CIFAR-10. This demonstrates that negative feedback is orthogonal to existing techniques in training GANs. The above results, as well as detailed settings and analysis, are added in Sec 6 and Appendix B in the revision.

[*1] Miyato, Takeru, et al. "Spectral Normalization for Generative Adversarial Networks." (2018).

---

### Author Response · Authors · 2019-11-15
**Looking forward to further feedbacks**

Dear Reviewers and AC:

Thank you again for your valuable and constructive comments, which are helpful for us.

We addressed all concerns for all reviewers and uploaded a revised version. We believe that our paper quality is improved. Specifically, we added further experimental results on SN-GAN with a significant improvement on previous state-of-the-art performance on CIFAR-10, which addressed the main concern for both Reviewer #1 and Reviewer #3.

So we sincerely appreciate it if the reviewers can take some time to return further feedbacks on whether our responses and new experiment results solve their concerns. If there is any other question, we will try our best to provide satisfactory answers.

Best,
the authors.

---

### Decision · Program_Chairs · 2019-12-19

**Decision:**

Reject

**Comment:**

This paper suggests stabilizing the training of GANs using ideas from control theory. The reviewers all noted that the approach was well-motivated and seemed convinced that that the problem was a worthwhile one. However, there were universal concerns about the comparisons with baselines and performance over previous works on Stabilizing GAN training and the authors were not able to properly address them.